# Serum-Based Proteomics Profiling in Adult Patients with Cystic Fibrosis

**DOI:** 10.3390/ijms21197415

**Published:** 2020-10-08

**Authors:** Hicham Benabdelkamel, Hanadi Alamri, Meshail Okla, Afshan Masood, Mai Abdel Jabar, Ibrahim O. Alanazi, Assim A. Alfadda, Imran Nizami, Majed Dasouki, Anas M. Abdel Rahman

**Affiliations:** 1Proteomics Resource Unit, Obesity Research Center, College of Medicine, King Saud University, P.O. Box 2925 (98), Riyadh 11461, Saudi Arabia; hbenabdelkamel@ksu.edu.sa (H.B.); afsmasood@ksu.edu.sa (A.M.); aalfadda@ksu.edu.sa (A.A.A.); 2Department of Biochemistry and Molecular Medicine, College of Medicine, Al Faisal University, Riyadh 11533, Saudi Arabia; hjalamri@alfaisal.edu; 3Department of Community Health Sciences, College of Applied Medical Sciences, King Saud University, 183T11, Riyadh 11495, Saudi Arabia; meokla@KSU.EDU.SA; 4Department of Genetics, King Faisal Specialist Hospital and Research Centre (KFSHRC), Zahrawi Street, Al Maather, Riyadh 11211, Saudi Arabia; Mai_Jabbar@kfshrc.edu.sa; 5The National Center for Biotechnology (NCB), Life Science and Environment Research Institute, King Abdulaziz City for Science and Technology (KACST), P.O. Box 6086, Riyadh 12354, Saudi Arabia; ialenazi@kacst.edu.sa; 6Department of Medicine, College of Medicine, King Saud University, P.O. Box 2925 (98), Riyadh 11461, Saudi Arabia; 7Lung Transplant Section, Organ Transplant Center, King Faisal Specialist Hospital and Research Center, Zahrawi Street, Al Maather, Riyadh 11211, Saudi Arabia; imranyaqoob@kfshrc.edu.sa; 8Department of Chemistry, Memorial University of Newfoundland, St. John’s, NL A1B 3X7, Canada

**Keywords:** cystic fibrosis, CFTR, proteomics, DIGE-MALDI/TOF, biomarker

## Abstract

Cystic fibrosis (CF), the most common lethal autosomal recessive disorder among Caucasians, is caused by mutations in the CF transmembrane conductance regulator (CFTR) chloride channel gene. Despite significant advances in the management of CF patients, novel disease-related biomarkers and therapies must be identified. We performed serum proteomics profiling in CF patients (*n* = 28) and healthy subjects (*n* = 10) using the 2D-DIGE MALDI-TOF proteomic approach. Out of a total of 198 proteins identified, 134 showed a statistically significant difference in abundance and a 1.5-fold change (ANOVA, *p* < 0.05), including 80 proteins with increased abundance and 54 proteins with decreased abundance in CF patients. A multiple reaction monitoring-mass spectrometry analysis of six differentially expressed proteins identified by a proteomic approach (DIGE-MALD-MS) showed a significant increase in C3 and CP proteins and a decrease in APOA1, Complement C1, Hp, and RBP4proteins compared with healthy controls. Fifteen proteins were identified as potential biomarkers for CF diagnosis. An ingenuity pathway analysis of the differentially regulated proteins indicates that the central nodes dysregulated in CF subjects involve pro-inflammatory cytokines, ERK1/2, and P38 MAPK, which are primarily involved in catalytic activities and metabolic processes. The involved canonical pathways include those related to FXR/RXR, LXR/RXR, acute phase response, IL12, nitric oxide, and reactive oxygen species in macrophages. Our data support the current efforts toward augmenting protease inhibitors in patients with CF. Perturbations in lipid and vitamin metabolism frequently observed in CF patients may be partly due to abnormalities in their transport mechanism.

## 1. Introduction

Cystic fibrosis (CF) is an autosomal recessive genetic disorder that causes significant morbidity and mortality, mainly in the Caucasian population [1,2,3]. It affects 1 in 2000–3000 newborns in the EU and 1 in 3500 in the USA [3]. The incidence of CF in Saudi Arabia has been estimated at 1 in 4243, and since there is no national screening program to identify CF in newborns, the average age at diagnosis is 2.8 ± 3.5 years [4].

Mutations cause CF in the CF transmembrane conductance regulator (CFTR) gene [5,6], which is a PKA-regulated chloride channel expressed in the apical membrane of the epithelial cells regulating secretion and absorption processes in tissues such as the lungs, pancreas, intestine, and kidneys, where it regulates water and salt balance [1,7,8]. Deletion of a phenylalanine residue at position 508 in the CFTR protein is the most frequent mutation that causes the CFTR protein to misfold, thereby remaining incompletely processed in the endoplasmic reticulum (ER), being subsequently degraded by an ER-associated degradation pathway [9,10,11]. Consequently, cells expressing the mutant protein are unable to transport chloride ions across the plasma membrane in response to a rise in intracellular cAMP levels [1]. Disruption of normal ion homeostasis in the airway causes mucoviscidosis (thick mucus), which makes the lungs more vulnerable to persistent and recurrent infections [5,6,12].

Chronic infections and persistent inflammation can deteriorate lung function and lead to a higher mortality rate in CF patients [2,6,12]. Dysfunctions in macrophages and neutrophils lead to increased chronic inflammation in CF patients. CFTR impairment in macrophages reduces their bacterial clearance abilities and induces pro-inflammatory cytokine production [6]. Moreover, CFTR knockout mice show annex-aggerated IgE response toward Aspergillus fumigatus with high levels of IL-13 and IL-4. T-cells appear to play a major role in disrupted immune responses because the lack of CFTR in CD3+lymphocytes results in aberrant cytokine secretion and adaptive immune responses [13]. In the CF lung, activation of the nuclear factor NF-κB signaling enhances the production of pro-inflammatory mediators, including interleukin8 (IL8), which is a potent neutrophil chemoattractant protein [12]. The chronic presence of neutrophils in the CF lung can cause irreversible damage to the lung parenchyma through the continued release of neutrophil proteases, particularly the neutrophil elastase [6] and ROS [12].

The newborn screening for CF is mainly performed by measuring the immunoreactive trypsinogen (IRT) on a dried blood spot (DBS), followed by CFTR DNA mutation screening [7,14]. Additionally, a few centers measure the levels of fecal elastase, which improves the diagnostic capability of CF newborn screening [15]. Confirmatory diagnosis is usually achieved by demonstrating the presence of two bi-parentally inherited CFTR mutations and abnormal sweat chloride excretions [16,17]. However, discovering new biomarkers to evaluate disease progression and therapeutic targets to reduce CF complications is still necessary. Proteomics has evolved as a promising platform for the identification of secreted proteins involved in CF pathophysiology, which could help in discovering new biomarkers or therapeutic targets for CF [2,12,18].

Several proteomic reports have shown differential protein expression in patients with CF. In one study, a total of 349 proteins related to CF pathogenesis and proteostasis of CFTR processing were differentially expressed in the bronchial epithelial cell models of CF [1]. In another report, ezrin, HSP70, endoplasmin, lamin A/C, and other CFTR-related proteins were identified as central hubs in CFTR homeostasis [5]. In primary cultures of human nasal epithelial cells obtained from CF patients with nasal polyps, proteomic analysis revealed changes in pathways related to metabolism, G-protein processing, inflammation, oxidative stress response, protein folding, proteolysis, and structural proteins [9]. An analysis of the sputum cellular proteins of CF patients with chronic Pseudomonas aeruginosa infection indicated the involvement of the Rho family of small GTPases, immune cell movement/activation, the generation of ROS, and the dysregulation of cell death and proliferation [18]. Another proteomic analysis of sputum from CF patients showed the differential expression of proteins related to proteolytic degradation and the influx of inflammation, such as myeloperoxidase, cleaved alpha-1-antitrypsin, IgG degradation, and IL8 [14].

Proteomic profiling has also been performed on the serum of patients with CF. Serum proteome profiling of CF patients with mild or severe respiratory diseases shows that CF perturbed pathways are related to lipid metabolism, platelets, and complement cascade activation. Furthermore, serum analysis indicates that the biological processes related to tissue destruction remodeling, protease/antiprotease balance, and innate immune function were all affected in CF patients [15]. In this study, we aimed to perform quantitative serum proteomics analyses in CF patients using the two-dimensional difference in gel electrophoresis (2D-DIGE) coupled with high-resolution mass spectrometry (MS) and to assess the differences compared with healthy subjects. Identifying differentially expressed proteins in CF patients may help with identifying novel candidate biomarkers associated with the disease and provide a deeper understanding of the pathological changes of CF.

## 2. Results 

### 2.1. Clinical Characteristics of Study Subjects

In this study, 28 young adult patients with a confirmed diagnosis of CF (based on clinical presentation, abnormal sweat chloride, and CFTR mutation analysis) and ten age-matched controls provided consent and blood samples for the proteomics analyses. Table 1 and Appendix A summarize the clinical, molecular, and routine laboratory findings in the CF patient cohort, which included 13 (46.4%) males and 15 (53.6%) females with a mean age of 20.3 ± 4.8 years (range 12–34). In this study, eight different CFTR mutations were identified, which represent 46.5% of all CFTR alleles reported in the Saudi Arabian patients with CF [19]. Eleven patients had deletions/frameshift mutations, nine shared single splicing (c.3700 A>G) mutations, and eight had missense mutations, all of which had been reported previously except the two mutations in patients CF3, CF6, and CF24. The c.3700 A>G is particularly interesting as it has been found to introduce a novel cryptic donor splicing site in exon 22 [20]. Therefore, this class V CFTR protein mutation is present in 32% (9/28) of CF patients, while class III and IV represented 50% and 18%, respectively. Only one CFTR mutated allele was identified in each of four patients (CF4, CF14, CF27, CF 34) in this cohort, indicating compound heterozygosity and non-consanguinity. Pancreatic insufficiency was present in 27 patients, and cystic fibrosis-related diabetes mellitus (CFRD) was reported in five patients. The mean body mass index (BMI, *n* = 28) was 18.4 ± 3.5. Serum/plasma total 25-hydroxy vitamin D (n = 28), vitamin E (*n* = 26), and IgE (*n* = 25) levels were also determined. The mean plasma total 25-hydroxyvitamin D level was 56 ± 23.8 nmol/L (reference: 13–76 nmol/L). One patient (CF2) had combined vitamin D and vitamin E insufficiency, while 4 patients (CF16, CF27, CF30, CF31) had elevated vitamin D levels. Vitamin E supplementation appears to be adequate as the mean serum vitamin E level was 11.7 ± 5.9 mg/L (reference: 5.5–15.5 mg/L). Total serum IgE level mean 282.4 ± 515.9 KU/L (reference: 5–500 KU/L). Low total serum IgE levels were found in two patients (CF2, CF26), while patients CF3 and CF34 had elevated levels. Patient CF11 underwent lung transplantation while patients CF6 and CF41 had received liver transplantation previously. Medications received by the patient cohort for standard clinical care included dornase alpha (recombinant DNAse1, all patients), hypertonic saline (CF1-5), tacrolimus, prednisone (CF11), and Ivacaftor (CF14).

A total of 1500 spots were mapped on the gels, of which 198 differed significantly (ANOVA *p* ≤ 0.05 and fold-change ≥ 1.5) between the CF and control samples (Appendix A).

The spot patterns were reproducible across all the gels, leading to alignment and further analysis. Cy2-labeling (the internal standard) was included to permit normalization across the complete set of gels and to allow a quantitative differential analysis of the protein levels. Significant changes in protein abundance levels were based on a one-way ANOVA (*p* ≤ 0.05 and fold-change ≥ 1.5 or <0.75). The statistical analysis using one-way ANOVA showed the expression of 198 protein spots to be significantly dysregulated between the study groups. The statistically significant spots (*n* = 198) were then manually excised from a preparative gel and identified using high-resolution mass spectrometry. 134 spots out of 198 were identified using a peptide mass fingerprinting (PMF) approach and SWISS-PROT database with a high degree of confidence (*p* < 0.05) (Table 2, Appendix A). The sequence coverage of the identified proteins by PMF ranged from 11% to 89%, with at least two peptides/proteins. In a few cases, variants of the same protein were detected in several spots on the gel (Appendix A, Table 2, Appendix A). Among the 134 identified entities, 80 proteins were up-regulated, and 54 were down-regulated in the CF patients compared with the controls. The up-regulated proteins included alpha-2-macroglobulin (A2MG), kininogen-1 (KNG1), alpha-1-antichymotrypsin (SERPINA3), alpha-1-antitrypsin (SERPINA1), coiled-coil domain-containing protein 68 (CCDC68), uromodulin (UMOD), complement factor H (CH), ceruloplasmin (CP), complement 3 (C3), and metabotropic glutamate receptor 4 (GRM4). The down-regulated proteins included type II inositol 1,4,5-trisphosphate (INPP5B), protein-glutamine gamma-glutamyltransferase6 (TGM6), RB1-inducible coiled-coil protein 1 (RB1CC1), putative protein FAM10A4 (ST13P4), dysferlin(DYSF), and lipoprotein lipase (LIPL). The complete list of up- and down-regulated proteins is shown in Table 2.

### 2.2. Mass Spectrometry Identification of Differentially Expressed Proteins on 2D-DIGE

The principal aim of this study was to perform a comparative serum proteome analysis of CF patients (*n* = 28) and healthy subjects (*n* = 10) to identify the differential expression patterns. Representative fluorescent protein profiles of 2D-DIGE containing the control subjects samples labeled with Cy3 are shown in Figure 1A, the CF patients’samples labeled with Cy5 (Figure 1B), the pooled internal control samples labeled with Cy2 (Figure 1C), and a merged 2D-DIGE comparison of Cy3/ Cy5 (D) are shown in Figure 1D.

### 2.3. Methodology Validation Using Multiple Reaction Monitoring (MRM) Mass Spectrometry

Six different significantly dysregulated proteins from the CF 2D DIGE proteomics profile were selected for validation. Signature peptides for these selected proteins were identified using the criteria that we described elsewhere [21]. The proteins were selected based on their involvement in the protein-protein interaction network pathway. Those proteins with a higher number of interactions and fold changes showing both an increased (Complement C3, Ceruloplasmin,) and decreased abundance (Apolipoprotein A-I, Haptoglobin, Complement C1 and Retinol-binding protein 4) were taken to confirm the findings. The uniqueness and reliability of these signature peptides were confirmed using SkeyLine Software V3 and PeptideAtlas and then synthesized as standard material (GeneMed Synthesis (San Francisco, CA, USA) [22]. An MRM method was developed using triple quadrupole mass spectrometry (LC-MSMS). Representative chromatograms of each protein’s signature peptide are shown in Appendix A. This validation experiment shows that these selected six proteins have similar expression trends in CF, as shown in Figure 2, with a different fold change value. The expression profiles of these proteins were statistically evaluated with unpaired t-tests using PrismPad Software.

### 2.4. Principal Component Analysis (PCA) and Cluster Analysis

The PCA performed on all spot features (*n* = 198) exhibited statistically significant (ANOVA *p* ≤ 0.05 and fold-change ≥ 1.5 or < 0.67) changes in abundance, as identified by MS. The PCA showed a separation between the two groups based on distinctive proteomics profiles (Appendix A). The differentially abundant spots showed expression pattern clusters according to their abundant patterns based on a hierarchical clustering analysis (Appendix A). The clustering pattern showed that the changes in the protein intensities for the selected spots differed significantly between the CF and control samples

### 2.5. Biomarker Evaluation and Analysis

The significantly expressed data from the previous analysis were further analyzed using the MetaboAnalyst software package, where the data were further normalized via a log transformation and Pareto scaling. The significant and dysregulated proteins in this study were further analyzed using univariate analysis and highlighted in a volcano plot, in which the x- and y-axes indicate the fold change (cutoff 1.5) and the y-axis t-test (FDR *p* < 0.05), respectively (Figure 3A). This rigorous analysis that combines expression fold change and FDR-corrected p-value (FDRp) showed that only 42 and 22 proteins were down- and up-regulated in the CF patients compared with the control patients, respectively. A multivariate statistical analysis was performed to analyze the serum proteomics dataset. The partial least squares discriminant analysis (PLS-DA) was first performed to visually reveal the distinct separation between the study groups (Q2: 0.805, and R2: 0.971) (Figure 3B).

The gel spot expression dataset was further analyzed using the biomarker evaluation feature of the MetaboAnalyst software, in which the PLS-DA model was used to generate an exploratory receiver operating characteristic (ROC) curve for the most frequent proteins as potential biomarkers (Figure 4A, Appendix A). The top 15 proteins show the highest potential to serve as a protein panel for identifying CF with a maximum area under the curve (AUC) of 0.981 with the number of latent being two. These 15 proteins are illustrated in Figure 4B based on their frequencies for being selected. The sample-set was divided into a test set, which contained 75% of the scans, and a validation set contained the remaining 25%. Representative ROC curves for a few potential biomarkers are displayed for apolipoprotein A-I (AUC 0.952) and protein-glutamine gamma-glutamyltransferase 6 (AUC 0.968) in Figure 4C,D, respectively.

### 2.6. Interactions of Identified Proteins and Network Connectivity Mapping Asing an Ingenuity Pathway Analysis (IPA)

The identified proteins in the dataset, which had significantly different abundances, were then probed using the IPA software tool. The IPA uses an algorithm that generates scores using the ingenuity knowledge database and attributes them to the corresponding best-fit biological network pathways. This connectivity map is enriched with other interacting nodes that are input by the software and the relationships are denoted as direct or indirect by continuous or discontinuous connecting lines, respectively. The network with the highest score between the CF and control samples was related to "metabolic diseases, neurological diseases, and disease organismal injury and abnormalities” (Figure 5A, Appendix A) and involved 16 proteins as focus molecules. Additionally, the top canonical pathway displaying significance was associated with FXR/RXR activation (Overlap = 8.8%, *p* = 2.12 × 10^−14^) (Figure 5B). The other canonical pathways are summarized in Appendix A.

### 2.7. Classification of Key Proteins Based On Function

Following the MS analysis, all successful 134 differentially abundant proteins identified between the CF and control samples were subjected to the PANTHER (protein analysis through evolutionary relationships) classification system (http://www.pantherdb.org/) according to their molecular function (Appendix A), biological process (Appendix A), and cellular component (Appendix A). The functional classification showed that most of the differentially expressed proteins were involved in catalytic activity (40.40%), followed by binding functions (30.0%). According to their biological processes, most (36.7%) of the identified proteins were involved in the metabolic process, followed by biological regulation (21.7%). Further, most of the identified proteins were cellular, whereas 20.5% were extracellular.

## 3. Discussion

In the present study, we compared the serum proteomic profiles between clinically stable CF patients (*n* = 28) and healthy subjects (*n* = 10) using the 2D-DIGE MALDI-TOF proteomics approach. Proteomic analysis of serum proteins may help elucidate the pathways and processes implicated in the disease to understand its pathophysiology better and may present opportunities for developing novel therapies or improved prognosis [12]. Overall, our results showed that a total of 134 proteins differed significantly between the CF and control samples, of which 80 were up-regulated and 54 were down-regulated in the CF patients. Several proteins (including Alpha-1-antitrypsin, Apolipoprotein A-I, ceruloplasmin, and alpha-2-macroglobulin) were found in more than one spot on different locations of the gel. The identification of the same proteins at different positions may correspond to their isoforms or their post-translational modifications as a result of acetylation, methylation, phosphorylation, or glycosylation that can either shift the protein right or left, depending on the isoelectric point (PI), or up and down depending on the modifications in molecular weight (MW). Our previous work has also reported the identification of such protein isoforms differing both in size and isoelectric point in plasma proteomics.

These differentially expressed proteins were functionally involved in regulating catalytic activities and metabolic processes related to the canonical pathways, including FXR/RXR and LXR/RXR activation, acute phase response signaling, IL12 signaling, and the production of nitric oxide and ROS (Figure 5A,B). The identified proteins were grouped according to their functions as follows: (i) proteases and antiproteases, (ii) complement system, (iii) redox status and antioxidants, (iv) mitochondrial proteins, (v) lipid metabolism, and (vi) vitamin transporters.

### 3.1. Proteases and Antiproteases

Proteases are involved in a series of intracellular and extracellular regulatory processes, including tissue remodeling, mucin expression, bacterial killing, and neutrophil chemotaxis. In the healthy lung, proteases and antiproteases maintain a homeostatic balance, preventing lung damage that may arise from proteases [6,18]. However, the CF lung is a protease-rich environment, and this protease burden overwhelms the antiprotease capabilities, leading to a protease-antiprotease imbalance, which is heavily implicated in CF pathophysiology [6,12]. A proteomic analysis of sputum collected from adults with CF with pulmonary exacerbation was characterized by extensive proteolytic degradation [14]. Furthermore, induced levels of 22 proteases and peptidases have been observed in human CF bronchoalveolar lavage fluid [12]. The up-regulation in the sputum protease matrix metalloproteinase9 (MMP9) has also been linked to reduced lung function and airway inflammation in children with CF [12].

Proteases are known to cleave cell-surface immune receptors, which interferes with the ability of monocytes or macrophages to recognize and clear infections and thus affects the immune systems of CF patients [5]. Neutrophil elastase (NE) is a major protease that actively contributes to lung damage in CF patients [6,23] through a vicious cycle involving NE up-regulates and pro-inflammatory cytokine secretions, leading to further neutrophil recruitment, which generates a continuous and destructive cycle of neutrophilic inflammation and protease release [6]. Compared with healthy controls, the sera of CF patients displayed a 4.77-fold increase in the expression of alpha-1-antitrypsin (A1AT/SERPINA1), which is an inhibitor of serine proteases with NE being its primary target. Consistent with our observation, previous reports showed that patients with CF secrete elevated levels of A1AT and other antiproteases [6]. Another antiprotease with a significant (3.43-fold) increase identified in our study was alpha-2-macroglobulin, which is known for its remarkable ability to inhibit a broad spectrum of antiprotease activities [24]. By contrast, we observed a decrease in antithrombin III, which is a serine protease inhibitor that regulates the blood coagulation cascade. Therapeutic strategies to boost levels of the protective antiproteases such as A1AT in the lungs remain a potentially attractive approach to protect the lungs from the damage caused by excess proteases in patients with CF [6].

### 3.2. Complement System

Progressive lung tissue damage in CF patients is mediated by a cycle of small airway obstruction, infection with microbial pathogens, and inflammation [23]. The complement proteins are key components of the innate immune system [16], which plays a critical role in the removal of pathogens and other dangerous particles, such as immune complexes, cellular debris, and dead cells [1,2,6]. As a result of infection, the activation of complement proteins leads to opsonization, phagocytosis, and destruction of the pathogen, initiation of inflammation, and finally activation of the adaptive immune response [16,25]. However, unregulated or persistent complement system activation triggers a destructive inflammatory cascade, which may lead to lung tissue damage and cause progressive loss of lung function [16,25].

Raised levels of pro-inflammatory complement proteins have been observed in the sputum of individuals with CF [16]. The proteomic analysis revealed that the complement proteins C3 and C4 are significant constituents of CF lung fluid. Furthermore, increased levels of C5a have been observed in the bronchoalveolar lavage fluid of CF patients with stable lung disease compared with healthy controls [23]. Dysregulation of the complement system in the CF bronchoalveolar lavage [fluid 12] or sputum [23] may impact lung disease pathogenesis [12] and correlate with clinical measures of CF disease [23], whereas IV antibiotic treatment of individuals with CF experiencing pulmonary exacerbations causes changes in several blood proteins involved in complement activation and inflammatory/immune-related pathways [16]. In our study, we found that CF patients had a 2.9-fold increase in complement C3 and a 3.41-fold increase in complement factor H, while the complement factor I was reduced by 1.5-fold compared with the control subjects.

### 3.3. Redox Status and Antioxidants

High levels of ROS and oxidative stress are characteristics of CF. Physiologically, CFTR controls intracellular ROS levels by regulating the intracellular and extracellular transport of glutathione, a major intracellular antioxidant protein. A defective CFTR, however, creates a state of redox imbalance leading to the generation of ROS, and a high degree of oxidative stress that is thought to contribute to reduced local and systemic levels of glutathione [26]. Glutathione homeostasis is highly regulated and maintained by the activity of the enzyme gamma-glutamyl transferase (GGT). GGT catalyzes the synthesis and breakdown of extracellular glutathione by salvaging the amino acids and maintaining levels of the rate limiting substrate cysteine, and through the cleavage of gamma-glutamyl peptide bonds transfers the gamma-glutamyl moiety to its acceptors [27,28]. In the present study, we found a decrease in the spots related to GGT in patients with CF compared with the controls. The decreased activity of this enzyme may reflect the heightened redox and oxidative state caused by CF. The decreased activity may also lead to altered levels of different metabolites, including glutamate, gamma-glutamyl linked amino acids, mercapturic acid, and cystine (unpublished data), which are intermediates in the synthesis of glutathione via the γ-glutamyl cycle. Increased cystine levels, the oxidized form of cysteine, is a known biomarker for both chronic fibrotic lung diseases and CF. The systemic deficiency of glutathione in CF has preempted its use clinically as a therapeutic agent in patients with CF for its mucolytic effects. The altered levels of this enzyme were assessed using the ROC curves, which demonstrated a heightened discriminatory value of the protein, making it a potential biomarker for detecting CF.

In the CF lung lavage fluid, the level of reduced glutathione, which acts as an antioxidant and in detoxification, was decreased. In CF mouse nasal epithelial cells, the levels of glutathione S-transferase (GST), which catalyzes the glutathione-mediated detoxification of oxidative stress products, peroxiredoxin 6 (Prdx6), a glutathione-dependent peroxidase (Gpx) involved in defense against oxidative stress, and Hsp27, a heat shock protein that increases intracellular levels of glutathione and acts as a chaperone for detoxification, were all reduced. Saline-induced sputum proteomic profiles from adults with CF and a pulmonary exacerbation were characterized by extensive proteolytic degradation and the influx of inflammation-related proteins, including myeloperoxidase, cleaved α1-antitrypsin, IgG degradation and n, IL8, and total protein concentration. Myeloperoxidase expression and IgG degradation were the strongest predictors of FEV1% [11]. Other proteomic studies have also identified the differential expression of myeloperoxidase, superoxide dismutase, catalase, and glutathione reductase in CF bronchoalveolar lavage fluid [12]. 

Other antioxidant proteins that demonstrated an increased abundance in our study include Ceruloplasmin a, which is a multi-copper oxidase that evolved to ensure the safe handling of free oxygen radical scavengers by binding molecular oxygen and reducing it to water. Ceruloplasmin is also an acute-phase reactant protein involved that participates in inflammatory responses and fluctuates significantly in several diseases and hormonal states [29]. In this study, we observed a significant increase in multiple protein spots relating to ceruloplasmin or its post-translational modifications in CF sera compared with healthy controls. Our findings are contrary to those of Charro et al. who, in their 2DE proteomics study, did not find any significant differences in its levels, although a tendency toward an increased level was noted in the nephelometric analysis [15]. However, we also observed a significant 3.15-fold reduction in spots relating to haptoglobin, which acts as an antioxidant and has antibacterial protein activity in CF serum, compared with healthy controls.

### 3.4. Mitochondrial Proteins

The differential expression of mitochondrial proteins has been reported in human CF nasal epithelial cells and bronchial tissue, implicating a CF-associated reduction in mitochondrial metabolism [12]. Cytochrome c oxidase (COX), the terminal enzyme of the mitochondrial respiratory chain, plays a crucial role in regulating mitochondrial energy production and cell survival. Such regulation ensures the building of highly efficient molecular machinery, which can catalyze the transfer of electrons from cytochrome c to molecular oxygen and ultimately to facilitate the aerobic production of ATP [30]. In our study, we observed a 2.27-fold decrease in the expression of the cytochrome c oxidase subunit 7A1 (COX7A1), a new member of the COX7A gene family, which is a subunit of the COX holoenzyme that is incorporated into the mitochondrial COX complex [24]. However, the specific mechanism(s) by which mitochondrial metabolism contributes to CF pathophysiology and the involvement of COX7A1 require further elucidation.

### 3.5. Lipid Metabolism

A major clinical feature shared by CF patients is exocrine pancreatic insufficiency, which results in visceral fat and multi-vitamin malabsorption and malnutrition and secondary essential fatty acid and fat-soluble vitamin deficiencies. However, fat malabsorption in CF patients may not be caused by pancreatic insufficiency alone, as noted by persistent steatorrhea, weak growth, and malnutrition, regardless of exogenous pancreatic enzyme supplementation. Therefore, persistent fat malabsorption in CF patients may also be because of abnormal plasma lipid transport [8]. In our study, we observed a 4.2-fold decrease in LPL expression, which plays an essential role in lipid clearance from the bloodstream, in lipid utilization, and in storage. We also observed a 2.16-fold decrease in apolipoprotein A1 (APOA1), which promotes cholesterol efflux from tissues. The biogenesis of APOA1 has been reported to be reduced in the intestinal tissue of CF patients. Moreover, the plasma concentrations of APOA1 were reduced in CF patients, likely due to the decline of APOA1 synthesis by the intestine [8]. A ROC analysis also showed the discriminatory capacity of APOA1 in cases of CF. However, further research is still needed to determine the role of plasma lipid transport in the pathophysiology of fat malabsorption in CF patients [8].

### 3.6. Vitamin Transporters

We observed a significant 2.7-fold decrease in the level of vitamin D-binding protein (GC, group-specific component, aka DBP), which is the major vitamin D-binding protein in plasma responsible for its transport and storage, in CF serum samples compared with healthy controls. Our findings are in line with those of Charro et al., who also showed a decrease in its these levels and attributed it to the poor nutritional status of patients with CF [15]. In addition, we observed a significant 2.25-fold reduction in retinol-binding protein 4 (RBP4), which is a specific transport protein for retinol that delivers retinol from the liver stores to the peripheral tissues, in CF serum samples compared with healthy controls. Likewise, murine CF airway epithelial cells showed a reduction in retinoic acid metabolism, which may implicate these transporters in the CF abnormal injury response, although their functional role has not yet been clarified [12].

### 3.7. Network Pathway Analysis

Bioinformatics analysis and IPA of the differentially regulated proteins indicated that the highest-scoring functional interaction network pathway identified in our study incorporated 13 focus proteins from our dataset that were related to metabolic diseases, neurological diseases, organismal injury, and abnormalities. The central nodes with the highest connectivity were found to be extracellular signal-regulated kinases (ERK), p38 mitogen-activated protein kinase (MAPK), and transforming growth factor signaling pathway. Additionally, proteins related to the involvement of pro-inflammatory cytokines were also observed, demonstrating the involvement of cytokines and the inflammatory pathway in patients with CF via the p38 MAPK pathway. p38MAPK is a Ser/Thr kinase that is critical in inflammation and the host response to stress signals such as those observed in CF. Previous studies have shown that the inhibition of p38 MAPK has the potential to reduce the inflammatory response and can have clinical and utility and be used therapeutically [31,32]. TGF-β1, as identified in the network pathways, is considered a crucial mediator of tissue fibrosis and causes tissue scarring. In CF, TGF-β1 has been described as causing mucociliary dysfunction and is known to down-regulate the expression and function of the CFTR protein when stimulated in human airway epithelial cells [33].

## 4. Materials and Methods

### 4.1. Ethical Considerations and Informed Consent

All procedures performed in this study involving human participants followed the ethical standards of the Declaration of Helsinki and the universal ICH-GCP guidelines. This study was reviewed and approved by the Institutional Review Board at King Faisal Specialist Hospital and Research Center (KFSHRC) (approval number 2160 031), Riyadh, Saudi Arabia. Written informed consent was obtained from all participants. This study was conducted at the Proteomics Unit, Obesity Research Center, College of Medicine, and the King Khalid University Hospital (KKUH), King Saud University, Riyadh, Saudi Arabia.

### 4.2. Study Design and Subjects

Adult individuals (28 CF patients and 10 control subjects) were enrolled in this study. The sample size was determined by conducting a power analysis using the Progenesis SameSpots Nonlinear Dynamics statistical software to determine the minimum number of required biological replicates (Appendix A). Blood samples were collected from adult patients diagnosed with CF who attended the adult CF-Pulmonology clinic at the King Faisal Specialist Hospital and Research Center (KFSHRC) (Riyadh, KSA). These patients were randomly selected during regular clinic visits and consented to participate in this study according to the Institutional Review Board (IRB) approval [34]. Detailed genetics and clinical baseline questionnaires were completed for each patient by the treating clinician (IN). Blood samples were collected using vein puncture into plain non-EDTA tubes (Vacutainer, BD Biosciences, San Jose, CA, USA). The serum was separated by centrifugation and was frozen immediately at −80 °C for further analysis. Any patient enrolled in another clinical study in the last 30 days, unable or unwilling to provide informed consent, or diagnosed with conditions other than CF was excluded from this study. The CFTR gene mutation analysis, DNA Isolation, PCR amplification of genomic DNA, mutational analysis, and sequencing methods have been described previously, and this data was collected from patients’ primary physicians (IN) [16].

### 4.3. Sample Processing and Protein Extraction

Thawed serum samples were centrifuged (5 min, 12,000× *g*), and the high-abundant plasma proteins (albumin, IgG) were depleted using an Albumin and IgG Depletion kit of 12-ml beads (Millipore, USA) according to the manufacturer’s instructions. The protein extraction was performed using 1,1,2-Trichloroethane (TCA)/acetone precipitation, as described by Chen et al [35,36].The protein concentration of each sample was then determined in triplicate using the 2D-Quant Kit (GE Healthcare, Chicago, IL, USA).

### 4.4. CyDye Labeling, 2D-DIGE, and Imaging

The proteins were labeled according to the manufacturer’s protocol (GE Healthcare, Chicago, IL, USA). Briefly, 50 µg of each CF and control protein extract sample was minimally labeled with 400 pmol of the N-hydroxysuccinimide esters of the Cy3 or Cy5 fluorescent cyanine dyes on ice for 30 min in the dark. A mixture of an equal amount of all samples was then pooled, labeled with Cy2, and used as an internal standard; this standard was normalized and matched across gels, dramatically decreasing gel-to-gel variation. A dye-switching strategy was applied during labeling to avoid dye-specific bias (Appendix A). First-dimension analytical gel electrophoresis was performed, followed by second-dimension sodium dodecyl sulfate-polyacrylamide gel electrophoresis (SDS-PAGE) on 12.5% fixed concentration gels, as previously described [36]. The gels were scanned with a Typhoon 9400 scanner (GE Healthcare, Chicago, IL, USA) using the appropriate wavelengths and filters for the Cy2, Cy3, and Cy5 dyes.

Differential in-gel electrophoresis (DIGE) images were analyzed using the Progenesis Same Spots v.3.3 software (Nonlinear Dynamics Ltd., UK). The gel images were first aligned together, and prominent spots were used to assign vectors to the digitized images within each gel manually. The automatic vector tool was next used to add additional vectors, which were manually revised and edited for correction if necessary. These vectors were used to warp and align gel images with a reference image of one internal standard across and within each gel. The gel groups were defined according to the experimental design, and the normalized volume of the spots was used to identify statistically significant differences. The software calculated the normalized volume of each spot on each gel from the Cy3 (or Cy5) to Cy2 spot volume ratio. The software performs a log transformation of the spot volumes to generate normally distributed data. The log normalized volume was used to quantify differential expression. Independent direct comparisons were made between the 28 CF patients and the 10 controls, and fold differences and p-values were calculated using a one-way ANOVA. All spots were pre-filtered and manually checked before applying the statistical criteria (ANOVA test, *p* ≤ 0.05, and fold difference ≥ 1.5). The normalized volume of spots, instead of spot intensities, was used in the statistical processing. Only those spots that fulfilled the above-mentioned statistical criteria were submitted for the MS analysis.

### 4.5. Protein Identification by MALDI-TOF MS

Coomassie-stained gel spots were excised manually, de-stained, washed, and digested according to previously described methods [36,37]. An aliquot of the digestion solution was mixed with an aliquot of a-cyano-4-hydroxycinnamic acid (BrukerDaltonics, Hamburg, Germany) in 30% aqueous acetonitrile and 0.1% trifluoroacetic acid. This mixture was deposited onto MALDI target plates (MTP 384 AnchorChip, 800 µm; BrukerDaltonics, Hamburg, Germany) and allowed to dry at room temperature. MALDI-MS(/MS) data were obtained using an Ultraflex time-of-flight (TOF) mass spectrometer equipped with a LIFT™-MS/MS device (ultrafleXtreme, BrukerDaltonics, Hamburg, Germany) as described previously [38,39,40]. A detailed analysis of the peptide mass mapping data was performed using Flex Analysis software v2.4 (BrukerDaltonics, Hamburg, Germany). MALDI-MS and MS/MS data were interpreted using BioTools v3.2 (BrukerDaltonics, Hamburg, Germany) in addition to the Mascot search algorithm (v2.0.04 updated 09/05/2018; Matrix Science Ltd., London, UK). Fixed cysteine modification with propionamide, variable modification due to methionine oxidation, one missed cleavage site (i.e., in case of incomplete trypsin hydrolysis), and a mass tolerance of 100 ppm were the primary MASCOT mass spectrometry peptide search criteria. Proteins were accepted as identification with a Mascot score higher than 56 and a *p*-value < 0.05. Not all spots of interest could be identified because some proteins were of low abundance and did not yield sufficiently intense mass fingerprints; other spots were mixtures of multiple proteins [41].

### 4.6. Multiple Reaction Monitoring-Tandem Mass Spectrometry for Validation

Six different proteins were selected from the proteomics profile of CF, and a signature peptide per protein was identified using the criteria that we described elsewhere [21]. These signature peptides were confirmed using SkeyLine Software V3 and synthesized for standard material (GeneMed Synthesis, San Francisco, CA, USA). A 2 mg/ml stock solution of each peptide was prepared in deionized water. Some peptides required pH adjustment to reach the maximum solubility (Appendix A). From the stock solution, 200 μg/mL working solutions were prepared in the mobile phase (90:10 ratio of 0.1% formic acid in H2O:0.1% formic acid in acetonitrile (ACN)) for mass spectrometric tuning and chromatographic optimization. The mass spectrometric transitions were developed after tuning using a Triple-Quadrupole-Tandem Mass spectrometer (XEVO TQD from Waters Corporation, Milford, CT, USA). Electrospray ionization (ESI) in a positive ionization mode was used while detecting the analytes. While infusing a peptide working solution, the precursor and product ions were monitored, including the collision and cone voltages, as summarized in Appendix A. The desolvation temperature and source temperature were set at 250 °C and 150 °C, respectively. The spraying gas flow rate was 500 L/hr, with a sample flow rate of 20 μL/min via a syringe infusion pump. The MS capillary source voltage was set at 1.98 KV, and the cone source voltage was set at 47 V. The optimum parameters for each peptide were used to establish the Multiple Reaction Monitoring (MRM) transitions for each analyte in the original MS method.

After constructing the MRM transitions, the targeted analytes were initially separated by reversed-phase chromatography, where the mobile phase gradient (90:10 H_2_O: ACN, 0.1% formic acid) was optimized for better baseline resolution and peak shape. A working solution of each analyte (200 μg/mL) was injected into Acquity Ultra Performance Liquid Chromatography (UPLC) C18, 1.7 μm, 2.1 × 50 mm columns at 25 °C. Eventually, the peptides were optimized for elution using a gradient fashion at a flow rate of 0.2 mL/min over a total of 10 min of run time. The gradient profile for solvent B (0.1% formic acid in ACN) was as follows: 10% for 1 min followed by a linear gradient to 90% over 5 min, which was then held at 90% for 0.2 min before returning to 10% for 0.3 min at 6.5 min post-injection. The column was equilibrated at 10% solvent B for 3.5 min before performing a second injection.

An AcquityUltra-High-Pressure Liquid Chromatography UPLC-XEVO TQD Triple-Quadrupole-Tandem Mass spectrometer (Waters Corporation, USA) was used to analyze the study samples. The eluted peptide from the chromatography was detected in the mass spectrometry-based on the optimum MRM after being ionized positively using ESI. The general MS parameters were modified for the LC-MSMS, where the desolvation temperature was set at 500 °C, the desolvation gas flow was set at 1000 L/Hr, the cone gas flow was set at 50 L/Hr, the MS capillary source voltage was set at 1.98 KV, and the cone source voltage was set at 47V. The total run time for each sample was 10 min at a mobile phase flow rate of 0.2 mL/min following the gradient table. The samples were stored in the autosampler at 4 °C, and the injection volume was 10 μL. Many intermediate washing steps were performed during the run to minimize any sample carryover.

### 4.7. Bioinformatic Analysis

Biomarker analysis of the proteomics expression profiles was performed using MetaboAnalyst version 3.0 (McGill University). The raw data were normalized to the sample total median to ensure that all samples were distributed normally. The proteomics differences among the study groups were corrected to make individual features more comparable by using log-transformation, and Pareto-scaling, respectively. Because the data were Gaussian-distributed, the unpaired two-tailed Student’s t-test was used for binary comparisons between any two study groups, where the significance levels for the protein data were considered at an The false discovery rate (FDR)-corrected *p* < 0.05 with a 1.5-fold cutoff change; the values are presented as the mean ± SEM. The ROC curves were constructed using a PLS-DA model from MetaboAnalyst software version 3.0 (McGill University, Montreal, Canada) (http://www.metaboanalyst.ca) for global analysis. The raw data were normalized, transformed, and median-, log-, and Pareto-scaled to ensure that all the data were visualized under the Gaussian distribution. Further analyses were performed on GraphPad Prism (version 6.0, Graph Pad Software, La Jolla, CA, USA).

The IPA version 9.0 (Ingenuity Systems, Redwood City, CA, USA) was used to analyze protein interaction networks, and the functions of the differentially expressed serum proteins in CF patients. IPA software maps the UniProt IDs into the Ingenuity Knowledge Base, which is the largest manually curetted resource combining information from all published scientific studies. This software aids in determining the functions and pathways that are most strongly associated with the MS-generated protein list by overlaying the experimental expression data onto networks constructed from published interactions.

## 5. Conclusions

Proteomic studies in CF have significantly improved our knowledge of CF and enhanced the understanding of its complex pathogenesis. Additional studies are necessary to identify novel biomarkers. In our study of serum samples from adult patients with CF, we identified 134 differentially expressed proteins by employing a 2D-DIGE MALDI-TOF proteomic approach. In general, proteins related to inflammation and tissue repairs, such as anti-proteases and complement factors, were perturbed in CF sera. Transport proteins of vitamin A and D and lipoproteins were down-regulated, suggesting possible explanations for their deficiencies in CF.

## Figures and Tables

**Figure 1 ijms-21-07415-f001:**
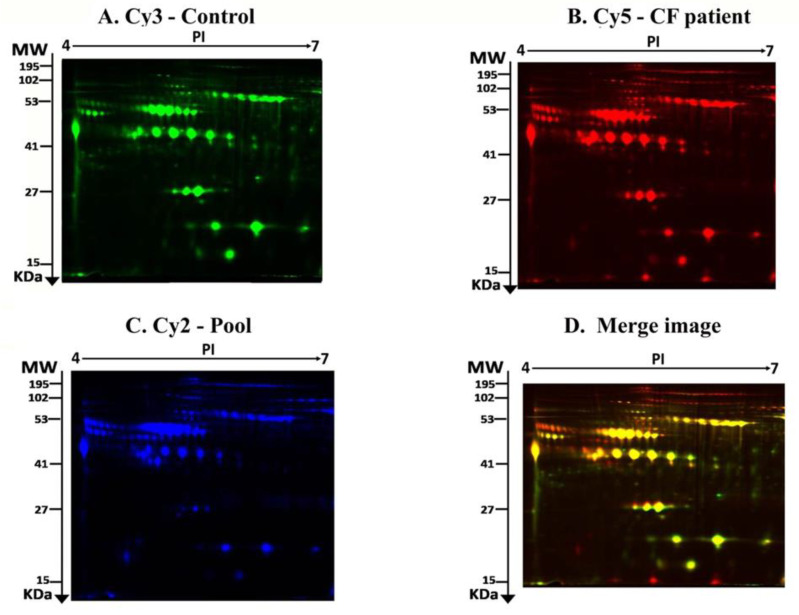
Representative fluorescent protein profiles of 2D-DIGE containing a control sample labeled with Cy3 (**A**), a CF sample labeled with Cy5, (**B**) a pooled internal control labeled with Cy2, (**C**), and a merged 2D-DIGE comparison Cy3/Cy5 (**D**).

**Figure 2 ijms-21-07415-f002:**
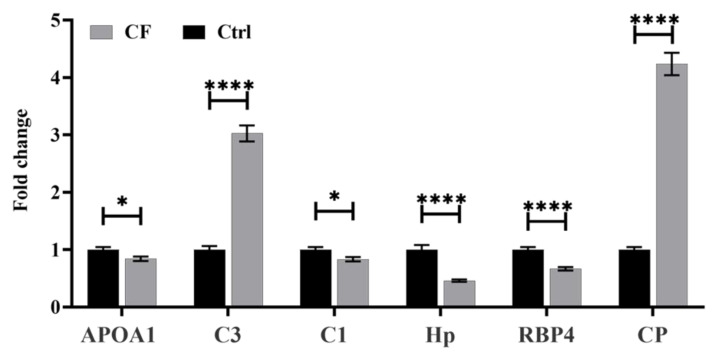
Multiple reaction monitoring (MRM)mass spectrometry for validating the study findings. The MRM method based on signature peptides was developed to validate the expression of six proteins found in the proteomics approach (DIGE-MALD-MS). The expression of these six proteins in CF patients was expressed in fold changes compared with the healthy controls (Ctrl). The statistical significance was evaluated using an unpaired t-test (*n* = 10), in which * represents *p* < 0.05, and **** represents *p* < 0.0001.

**Figure 3 ijms-21-07415-f003:**
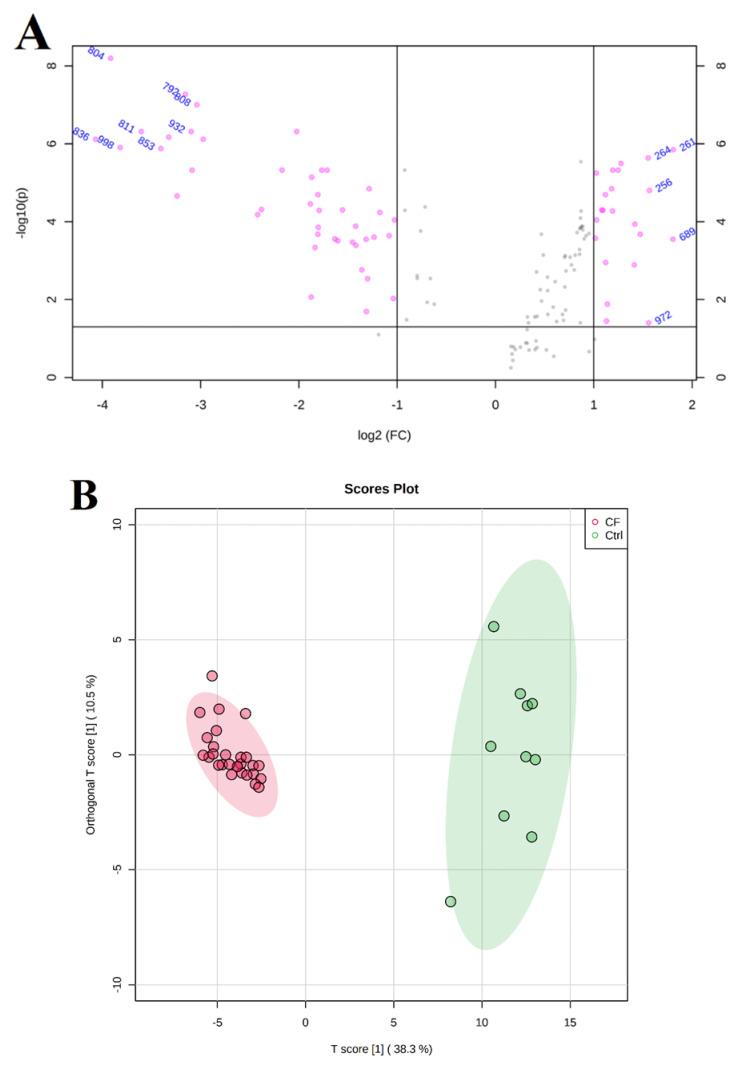
Statistical analysis of proteomics expression for the CF patients compared with the healthy control subjects. A volcano plot between the Ctrl and CF groups shows the significantly dysregulated proteins (42 down-regulated and 22 up-regulated in the CF group), with cutoffs of 2 and 0.05 for the fold change (x-axes) and t-test, respectively (**A**). An orthogonal PLS-DA score plot with eight components indicates a significant separation between the study groups (Q2: 0.805, and R2: 0.971) for 1000 permutations due to the proteomics dysregulations (**B**).

**Figure 4 ijms-21-07415-f004:**
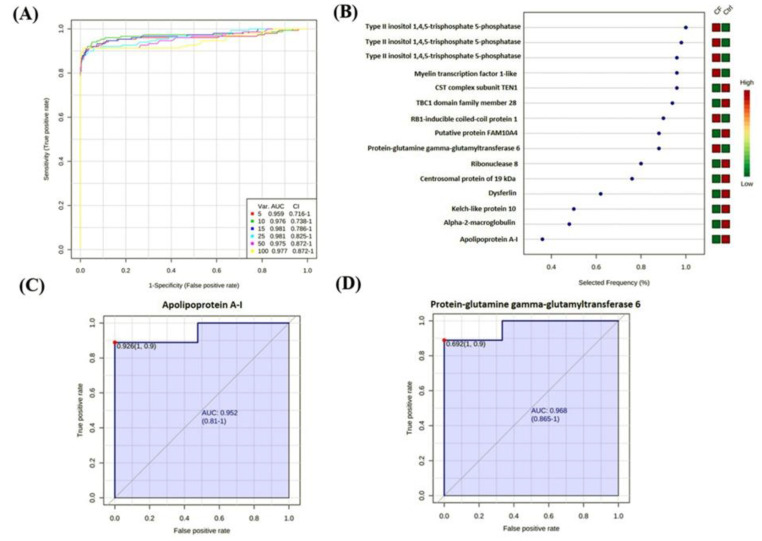
Biomarker statistical evaluation. An ROC exploratory analysis was generated by the PLS-DA model with many latent variable 2 and with increased sensitivity (x-axis) and specificity (y-axis), in which the area under the curve is at least 0.959 with a minimum combination of five variables (**A**). The features are ranked based on the selected frequency using the PLS-DA model, in which the color change from green to red indicates their relative expression of low to high, respectively (**B**); represents isoforms of the same protein found in different spots of the gel. Representative ROC curves for apolipoprotein A-I, which is down-regulated in CF (**C**), and protein-glutamine gamma-glutamyltransferase 6, which is up-regulated in CF (**D**).

**Figure 5 ijms-21-07415-f005:**
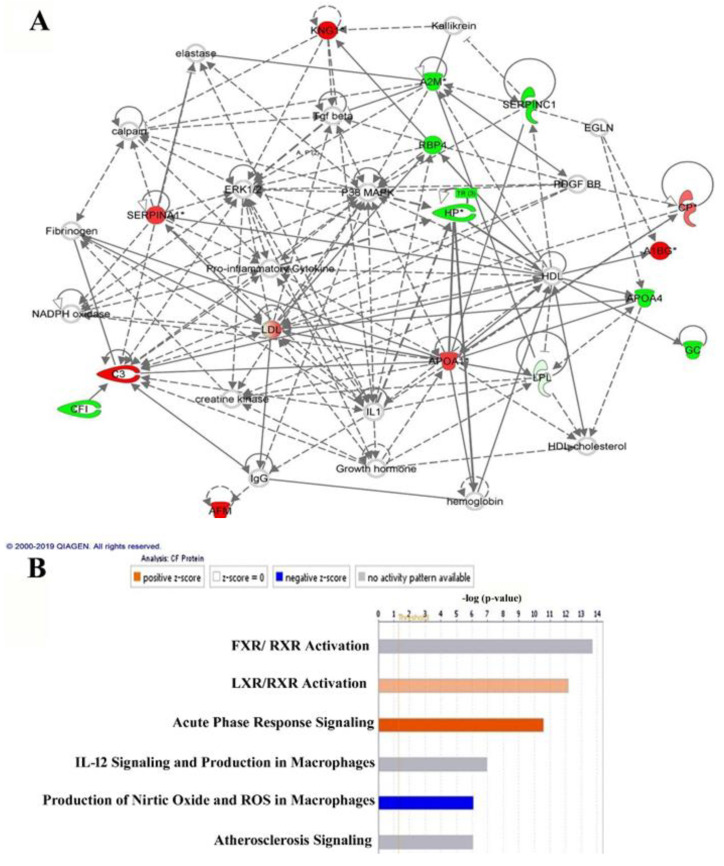
Schematic representation of the most significant IPA networks involving the proteins that were differentially regulated between the CF and control states. The IPA analysis found that the functional interaction networks pathway with the highest score (33) was related to “metabolic diseases, neurological diseases, and disease organismal injury and abnormalities.” This pathway incorporated pro-inflammatory cytokines, ERK1/2, and P38 MAPK as central nodes that were deregulated in CF samples. The nodes in green and red correspond to down-regulated and up-regulated proteins, respectively. The colorless nodes were proposed by the IPA and suggest potential targets that are functionally coordinated with the differentially abundant proteins (**A**). The solid lines indicate direct molecular interactions, and the dashed lines represent indirect interactions. The diagram shows the top six canonical pathways ranked by the P-values obtained from the IPA (**B**).

**Table 1 ijms-21-07415-t001:** Clinical, biochemical and molecular genetics characteristics parameters of the CF patients in this study.

Sample ID	Age (years)	Gender	*CFTR* Mutation *	CFTR Protein Mutation Class	Sweat Chloride (meq/L)	FEV1 L (%)	Pancreatic Status
**CF1**	23	M	exon 12: c.1647 T>G; p.S549R	class III	NA	3.03(70%)	PI
**CF2**	22	F	exon 11: c.1418delG; p.G473EfsX54	class III	NA	0.47(15%)	PI
**CF3**	19	F	exon 15: c.2619 G>C); p.E873D	class III	105	1.17 (46%)	PI
**CF4**	20	M	exon 14: c.1911 delG; p.Q637HfsX26 (het)	class III	110	1.68(35%)	PI
**CF5**	22	F	exon 22: c.3700 A>G; p.I1234V	class V	100	0.92(31%)	PI
**CF6**	17	F	exon11:c.1416 delG; p.M472fs	class III	NA	1.23(41%)	PI
**CF7**	22	M	exon 11: c.1418 delG; p.G473EfsX54	class III	79	1.12(31%)	PI
**CF9**	18	F	exon 11: c.1418 delG; p.G473EfsX54	class III	NA	2.08(67%)	PI
**CF10**	28	M	exon 4: c.416 A>T; p.H139L	class IV	NA	0.6(16%)	PI
**CF11**	26	M	exon 22: c.3700 A>G; p.I1234V	class V	NA	NA	PI
**CF12**	34	F	exon 11: c.1418del G; p.G473EfsX54/Exon 12: c.1736 A>G; p.D579G	class III	58	1.23(45%)	PI
**CF13**	20	M	exon 22: c.3700 A>G; p.I1234V	class V	108	NA	PI
**CF14**	17	M	exon 12: c.1647 T>G; p.S549R (het)	class III	124	3.07(84%)	PI
**CF16**	18	F	exon12:c.1647T>G; p.S549R	class III	NA	2.56(85%)	PI
**CF22**	14	F	exon 4: c.416 A>T; p.H139L	class IV	NA	0.68(24.9%)	PI
**CF24**	17	F	exon 13: c.2043 delG	class III	NA	2.32(84%)	PI
**CF25**	20	F	exon 11: c.1418 delG; p.G473EfsX54	class III	NA	1.97(70%)	PI
**CF26**	26	M	exon 22: c.3700 A>G; p.I1234V	class V	NA	0.99 (27%)	PI
**CF27**	14	M	exon11:c.1416 delG; p.M472fs (het)	class III	NA	2.47(64%)	PI
**CF29**	26	F	exon 22: c.3700 A>G; p.I1234V	class V	60	2.71(88%, post TX)	PI
**CF30**	12	M	exon 22: c.3700 A>G; p.I1234V	class V	88	2.58(81%)	PI
**CF31**	24	F	exon 22: c.3700 A>G; p.I1234V	class V	NA	1.68(61%)	PI
**CF32**	20	F	exon 22: c.3700 A>G; p.I1234V	class V	NA	1.3(47%)	PI
**CF33**	22	F	exon 22: c.3700 A>G; p.I1234V	Class V	NA	NA	NA
**CF34**	18	M	exon 11: c.1416 delG; p.M472fs (het)	class III	81	0.95(24%)	PI
**CF40**	18	F	exon4:c.416A>T; p.H139L	class IV	NA	0.63(16%)	PI
**CF41**	14	M	exon 4: c.416 A>T; p.H139L	class IV	NA	1.57(48%)	PI
**CF42**	18	M	exon 19-21 del	class IV	NA	3.08(77%)	PI

* All CFTR mutations are homozygous except for patient CF12 who was a compound heterozygote. In patients C4, C14, C27, and C34, only a single CFTR mutation was identifiable. Abbreviation: CFTR: cystic fibrosis transmembrane conductance regulator; FEV1: forced expiratory volume in second; het: heterozygous; L: liter; NA: not available; PI: pancreatic insufficient; PS: pancreatic sufficient; Tx: transplantation).

**Table 2 ijms-21-07415-t002:** List of the identified proteins (*n* = 134) using high-resolution mass spectrometry-based on fold change differential expression between the CF and control samples. The table shows the average ratio values for the CF and control patients, with their corresponding levels of fold change and one-way ANOVA (*p* < 0.05) using 2D-DIGE. [Analysis type: MALDI-TOF; database: SwissProt; taxonomy: homo sapiens].

Spot No	Accession No	Protein Name	*p*-Value (ANOVA)	CF/C Ratio	EXP
804	P32019	Type II inositol 1,4,5-trisphosphate	7.28 × 10^−14^	−13.06	Down
998	Q9UL68	Myelin transcription factor 1-like	1.57 × 10^−7^	−9.94	Down
811	P32019	Type II inositol 1,4,5-trisphosphate	2.54 × 10^−9^	−9.45	Down
853	Q8TDY2	RB1-inducible coiled-coil protein 1	5.48 × 10^−11^	−9.22	Down
796	Q8IZP2	Putative protein FAM10A4	7.08 × 10^−11^	−8.3	Down
987	Q86WV5	CST complex subunit TEN1	2.75 × 10^−7^	−7.35	Down
792	O95932	Protein-glutamine gamma-glutamyltransferase 6	5.04 × 10^−12^	−7.23	Down
932	Q8TDE3	Ribonuclease 8	1.94 × 10^−9^	−6.95	Down
848	Q96LK0	Centrosomal protein of 19 kDa	4.05 × 10^−8^	−6.77	Down
787	O75923	Dysferlin	9.09 × 10^−10^	−6.4	Down
808	Q2M2D7	TBC1 domain family member 28	1.94 × 10^−9^	−6.21	Down
700	P01877	Immunoglobulin heavy constant alpha 2	7.09 × 10^−4^	−5.6	Down
967	Q9UL68	Myelin transcription factor 1-like protein	8.76 × 10^−6^	−4.2	Down
823	P06858	Lipoprotein lipase	1.15 × 10^−9^	−4.2	Down
863	Q6UWF9	Protein FAM180A	1.87 × 10^−6^	−3.9	Down
907	P02647	Apolipoprotein A-I	2.95 × 10^−6^	−3.36	Down
782	P00738	Haptoglobin	6.59 × 10^−5^	−3.15	Down
834	O60869	Endothelial differentiation-related factor 1	4.43 × 10^−7^	−3.1	Down
714	Q96QP1	Alpha-protein kinase 1	2.02 × 10^−5^	−2.98	Down
730	Q6JEL2	Kelch-like protein 10	1.15 × 10^−5^	−2.97	Down
966	P00738	Haptoglobin	0.005	−2.94	Down
591	O00487	26S proteasome non-ATPase regulatory subunit 14	6.53 × 10^−5^	−2.82	Down
857	Q9UL68	Myelin transcription factor 1-like protein	6.76 × 10^−7^	−2.82	Down
763	P62906	60S ribosomal protein L10a	7.13 × 10^−5^	−2.77	Down
785	P02774	Vitamin D-binding protein, DBP	2.61 × 10-5	−2.7	Down
856	Q49AM3	Tetratricopeptide repeat protein 31	3.24 × 10^−8^	−2.64	Down
789	Q99570	Phosphoinositide 3-kinase regulatory subunit 4	1.09 × 10^−6^	−2.55	Down
570	P02768	Serum albumin	8.38 × 10^−6^	−2.47	Down
893	P02647	Apolipoprotein A-I	1.27 × 10^−6^	−2.45	Down
829	P24310	Cytochrome c oxidase subunit 7A1	1.52 × 10^−4^	−2.27	Down
919	P02753	Retinol-binding protein 4	7.97 × 10^−5^	−2.25	Down
900	P02647	Apolipoprotein A-I	8.06 × 10^−4^	−2.16	Down
589	P02768	Serum albumin	9.00 × 10^−5^	−2.16	Down
1012	Q6ZN57	Zinc finger protein 2 homolog, Zfp-2	0.002	−2.13	Down
982	Q8TAA9	Vang-like protein 1	0.013	−2.09	Down
840	O95922	Probable tubulin polyglutamylase TTLL1	0.004	−2.06	Down
820	P02647	Apolipoprotein A-I	8.46 × 10^−4^	−2.02	Down
567	Q96PX9	Pleckstrin	3.10 × 10^−6^	−2.02	Down
862	P24310	Cytochrome c oxidase subunit 7A1	0.044	−1.97	Down
759	P00738	Haptoglobin	0.005	−1.95	Down
801	Q96SI1	BTB/POZ domain-containing protein KCTD15	9.35 × 10^−5^	−1.88	Down
839	P31751	RAC-beta serine/threonine-protein kinase	0.003	−1.86	Down
709	Q8TDE3	Ribonuclease 8	1.61 × 10^−4^	−1.86	Down
710	P01008	Antithrombin-III	0.011	−1.8	Down
896	P02647	Apolipoprotein A-I	0.002	−1.66	Down
668	Q9BQ50	Three prime repair exonuclease 2	2.32 × 10^−4^	−1.62	Down
604	P02790	Hemopexin	0.017	−1.5	Down
473	P24666	Low molecular weight phosphotyrosine protein phosphatase	0.002	−1.5	Down
773	P05156	Complement factor I	7.98 × 10^−4^	−1.5	Down
895	P02647	Apolipoprotein A-I,	0.026	−1.35	Down
598	P02790	Hemopexin	0.028	−1.26	Down
566	P04217	Alpha-1B-glycoprotein	0.02	−1.24	Down
972	Q9NR11	Zinc finger protein 302	3.53 × 10^−4^	4.94	Up
689	P01009	Alpha-1-antitrypsin	1.97 × 10^−6^	4.77	Up
908	Q9NT22	EMILIN-3	0.001	4.49	Up
261	P01023	Alpha-2-macroglobulin	2.35 × 10^−8^	4.45	Up
916	P01042	Kininogen-1	0.004	4.3	Up
264	P01023	Alpha-2-macroglobulin	3.13 × 10^−8^	3.786	Up
688	P01009	Alpha-1-antitrypsin	5.99 × 10^−6^	3.78	Up
256	P01023	Alpha-2-macroglobulin	7.45 × 10^−7^	3.68	Up
596	P00450	Ceruloplasmin	2.96 × 10^−5^	3.67	Up
695	P01009	Alpha-1-antitrypsin	2.16 × 10^−6^	3.6	Up
184	P01023	Alpha-2-macroglobulin	1.06 × 10^−9^	3.43	Up
270	P08603	Complement factor H	5.71 × 10^−6^	3.41	Up
163	P01023	Alpha-2-macroglobulin	3.48 × 10^−9^	3.35	Up
243	P01023	Alpha-2-macroglobulin	5.45 × 10^−9^	3.23	Up
234	P01023	Alpha-2-macroglobulin	1.06 × 10^−6^	2.97	Up
771	P01024	Complement C3	8.69 × 10^−6^	2.9	Up
260	P01023	Alpha-2-macroglobulin	2.21 × 10^−7^	2.85	Up
614	P01011	Alpha-1-antichymotrypsin	2.30 × 10^−8^	2.85	Up
225	P01023	Alpha-2-macroglobulin	6.36 × 10^−7^	2.8	Up
273	P01023	Alpha-2-macroglobulin	3.16 × 10^−7^	2.8	Up
247	P01023	Alpha-2-macroglobulin	5.45 × 10^−9^	2.74	Up
291	P01023	Alpha-2-macroglobulin	2.00 × 10^−6^	2.7	Up
693	P01009	Alpha-1-antitrypsin	0.001	2.68	Up
277	P01023	Alpha-2-macroglobulin	1.31 × 10^−6^	2.67	Up
294	P01023	Alpha-2-macroglobulin	1.37 × 10^−6^	2.66	Up
592	P15622	Zinc finger protein 250	3.30 × 10^−4^	2.65	Up
766	Q6UXP9	Putative uncharacterized protein	0.004	2.6	Up
283	P01023	Alpha-2-macroglobulin	3.64 × 10^−7^	2.58	Up
150	Q9H2F9	Coiled-coil domain-containing protein 68	1.19 × 10^−6^	2.54	Up
259	P01023	Alpha-2-macroglobulin	9.20 × 10^−7^	2.53	Up
276	P01023	Alpha-2-macroglobulin	3.98 × 10^−7^	2.51	Up
281	P01023	Alpha-2-macroglobulin	2.08 × 10^−6^	2.5	Up
601	P00450	Ceruloplasmin	0.003	2.43	Up
285	P01023	Alpha-2-macroglobulin	1.97 × 10^−6^	2.4	Up
266	P01023	Alpha-2-macroglobulin	6.86 × 10^−5^	2.38	Up
87	P01023	Alpha-2-macroglobulin	2.58 × 10^−6^	2.34	Up
635	P01042	Kininogen-1	4.46 × 10^−9^	2.33	Up
289	P01023	Alpha-2-macroglobulin	1.67 × 10^−6^	2.3	Up
149	P01023	Alpha-2-macroglobulin	9.97 × 10^−7^	2.3	Up
334	P07911	Uromodulin	3.79 × 10^−6^	2.25	Up
610	P01011	Alpha-1-antichymotrypsin	1.65 × 10^−6^	2.23	Up
367	P00450	Ceruloplasmin	9.27 × 10^−5^	2.18	Up
255	P01023	Alpha-2-macroglobulin	8.08 × 10^−6^	2.15	Up
253	P01023	Alpha-2-macroglobulin	3.64 × 10^−6^	2.12	Up
331	P00450	Ceruloplasmin	7.60 × 10^−6^	2.09	Up
338	P00450	Ceruloplasmin	1.83 × 10^−5^	2.07	Up
611	Q9H0J9	Poly [ADP-ribose] polymerase 12	0.003	2.06	Up
326	P00450	Ceruloplasmin	5.77 × 10^−6^	2.05	Up
362	P01023	Alpha-2-macroglobulin	2.62 × 10^−5^	2.03	Up
374	P01023	Alpha-2-macroglobulin	0.002	2.01	Up
388	Q14833	Metabotropic glutamate receptor 4, mGluR4	0.002	1.96	Up
377	P01023	Alpha-2-macroglobulin	0.005	1.93	Up
257	P01023	Alpha-2-macroglobulin	4.37 × 10^−5^	1.93	Up
376	P00450	Ceruloplasmin	0.004	1.9	Up
638	P01042	Kininogen-1	7.75 × 10^−6^	1.9	Up
405	P00450	Ceruloplasmin	4.61 × 10^−4^	1.89	Up
389	P01023	Alpha-2-macroglobulin	6.16 × 10^−4^	1.88	Up
332	Q14833	Metabotropic glutamate receptor 4	1.69 × 10^−4^	1.85	Up
263	P01023	Alpha-2-macroglobulin	7.51 × 10^−5^	1.82	Up
408	P43652	Afamin	2.83 × 10^−4^	1.76	Up
460	P00450	Ceruloplasmin	0.011	1.75	Up
949	Q8TDE3	Ribonuclease 8	0.017	1.74	UP
651	P01009	Alpha-1-antitrypsin	1.62 × 10^−5^	1.7	Up
645	P01009	Alpha-1-antitrypsin	1.72 × 10^−4^	1.69	Up
641	P01009	Alpha-1-antitrypsin	0.003	1.68	Up
343	P00450	Ceruloplasmin	8.46 × 10^−4^	1.68	Up
659	P01009	Alpha-1-antitrypsin	1.26 × 10^−5^	1.68	Up
345	P00450	Ceruloplasmin	4.95 × 10^−4^	1.67	Up
656	P01009	Alpha-1-antitrypsin	1.08 × 10^−5^	1.67	Up
339	P00450	Ceruloplasmin	9.42 × 10^−5^	1.66	Up
649	P01009	Alpha-1-antitrypsin	3.34 × 10^−5^	1.65	Up
650	P01009	Alpha-1-antitrypsin	4.28 × 10^−4^	1.61	Up
642	P01009	Alpha-1-antitrypsin	0.007	1.59	Up
351	Q8NDZ2	SUMO-interacting motif-containing protein 1	0.007	1.55	Up
463	Q8TCP9	Protein FAM200A	0.007	1.51	Up
502	P04217	Alpha-1B-glycoprotein	0.002	1.51	Up
560	P04217	Alpha-1B-glycoprotein	0.008	1.5	Up
558	Q9NXU5	ADP-ribosylation factor-like protein 15	0.003	1.5	Up
404	P00450	Ceruloplasmin	0.022	1.47	Up
465	P04217	Alpha-1B-glycoprotein	0.02	1.37	Up
559	P00450	Ceruloplasmin	0.01	1.36	Up

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
