# Peer review of "Serum-Based Proteomics Profiling in Adult Patients with Cystic Fibrosis"

_ijms, 2020, doi:10.3390/ijms21197415_

Round 1
Reviewer 1 Report
I am delighted with the revised manuscript.
Author Response
Thank you for your positive response and accepting our response and modifications
Reviewer 2 Report
Comments to the Authors
The paper entitled “Serum-based proteomics profiling in adult patients with cystic fibrosis,” from Benabdelkamel H et al characterize the proteomic differences in plasma samples from patients with CF compared to a control group from subjects without CF.
This work represents an interesting aspect of CF research, usage of plasma samples from patients with CF and controls to discriminate by proteomics the proteins up and downregulated in the comparison between the two groups. They perform the proteomics analysis on the samples and then analyse the results using several bioinformatic tools. The research seems well constructed but further explanations of results from MS are required and the manuscripts needs to be re-structured in order for the reader to be able to better understand results and the conclusions behind those results. In summary, this paper brings improve of knowledge on the differences between CF and non-CF subject that may help find new biomarkers in the CF field, however needs major reformulation before be considered to publication.
There are several concerns that need be addressed before the manuscript can be considered for publication.
Major concerns:
The results start with the characteristic of the patients with CF. However, is not clear is the identification of the CFTR mutations was done in the study or just collect from the clinical info from the patients already existent.
. Title of the table is biochemical and genetic parameters but only genetic info is there. What do the authors consider biochemical? The mutation info in table 1, is confused, being the definition of the mutations is messy. The name of the mutations is giving in two different forms of nomenclature: the protein definition (amino acid alteration) and the cDNA/RNA definition (nucleic acid alteration). However, only one mutation is given per patient in the table, normally the genotype of patients with CF is compose of two mutations, one in each allele. Thus, it is not clear if the patients only had one mutation identified. If not and two mutations are identified why only one is referred in the table?
For the characterization of the CF label, the genotype (2 CFTR mutations) should be given (only one name is necessary, name the description of the nucleotide change is better), the sweat test value should also be given (since it is the CF standard diagnostic test), the FEV1 if available should also be given and pancreatic status could also be in Table 1 instead of table S1. This information will help understand the CF diagnostic and the severity of the disease. The IVS9 polymorphisms info can go to the supplementary table since no 5T was identified and the other ones are not considered CF causing variants.
Figure 2, should go to supplementary material, it is too crowded with number making it impossible to read. In the main manuscript, figure 2 should be presented the gel with only the 198 spots found differentially expressed numbered, in order to be easier to check this information and visually more understandable.
Table 2 should present the proteins ordered by fold changes from higher to lower, and with upregulated and down regulated protein separated. This allows an easier consultation of the table and a better understanding of the results.
One major concern about the results is the fact that many proteins correspond to more then one spot, being the alpha-2-macroglobulin represented by 30 different spots (since the numbers are not easy to identify in the gel it is difficult to know if all this 30 spot are closed in the gels or in completely different positions). How do the authors explain this phenomenon? Post-translation modification can not explain all these 30 different spots identified as the same protein. Alpha-1-antitripsin (7 spots) and Apolipoprotein A-I (6X) are also over represented in the gel. Is contamination during the preparation of the spots for the MS identification a possibility? Can post-translation modification and protein cleavage be confirmed by extra analysis to verify if some of these extra spots identified as the same protein are really coming from different the post-translation modifications? It is strange to have 30 different forms of a specific protein and need more explanation from the authors since all the results and conclusion depend on the accuracy of the identification of the proteins present in the gel.
Based on the PI and molecular mass know theoretically for each protein, is it possible to infer which would correspond to the active protein (know form of that protein) in the graph? In the text when the authors refer to a specific protein fold change in the plasm from the two groups they always use the results of the spot with the highest fold change (even if there are 30 different spots) how is it possible to know that that is the active form of the protein in the subjects?
Is the putative localization (based on theoratical pi and MW) is known the authors can verify which spot would be the best candidate for the know form of that protein and verify which is the fold change of that spot.
The last sentence in line 186,187 and 188 should go to the discussion section, since it does not correspond to results but to an explanation about the results.
What do the authors mean by non-human expressed protein (line 188)? The samples are from humans so what does that mean?
In figure 3, why were those 6 proteins chosen to further analysis? Do they correspond to the ones with higher fold changes of graph in figure 2? The reason for this selection is not clear in the manuscript.
In 2.3 section line 177 it is stated that from the 134 spots identified 80 were upregulated and 54 were downregulated in CF compared to controls. However, in 2.5 section line 230 using another methodology, 42 proteins were found down regulated and 22 upregulated in CF compared to controls. How do the authors explain that in one analysis gives more upregulated protein in CF compared to controls and another gives more downregulated protein in CF compared to controls?
Figure 4B there are problems with the numbers in the graph that are too big covering the legend og x-axis. In this graph it appears that crtl samples (light circles are in the) are in the separated group identified in the graph as CF group on the left, what are the authors comment on this effect?
The discussion is too extensive and mainly concerning figure 6 results. The authors should also discuss the other results like the explanation why so many spots are identified as one single protein. Why they select the protein present in figure 3. And discuss about biomarkers that were selected in the manuscript.
Furthermore, each discussion section starts with a theoretical introduction, being sometimes not so clear the connection to the results. The authors should first refer the results that they intent to discussion and then do the correlation with bibliography.
Minor concerns:
In the introduction it is stated that CFTR is present (line 67) “…..in apical membrane of secretory epithelial cells”, however CFTR can do both secretion and absorption.
In line 71 it is stated “…(ER), where it is subsequently degraded.” However, CFTR is not degraded in the ER.
Spaces are lacking between several words throughout the manuscript, this need to be checked and altered.
Round 2
Reviewer 2 Report
Comments to the Authors_review 2
The paper entitled “Serum-based proteomics profiling in adult patients with cystic fibrosis,” from Benabdelkamel H et al characterize the proteomic differences in plasma samples from patients with CF compared to a control group from subjects without CF.
This work represents an interesting aspect of CF research, usage of plasma samples from patients with CF and controls to discriminate by proteomics the proteins up and downregulated in the comparison between the two groups. They perform the proteomics analysis on the samples and then analyse the results using several bioinformatic tools. In summary, this paper brings improve of knowledge on the differences between CF and non-CF subject that may help find new biomarkers in the CF field, however needs major reformulation before be considered to publication. The research seems well constructed and some of my concerns are already addressed in the first review however there are still some issues that need to be clarified before the paper can be considered for publication.
Major concerns:
- From the info given by the authors to my first comment about mutations it is possible to infer that these results were not obtained during this study, just collected from the patient info by the clinician. Thus, sub-section 2.1 and table 1 should not be included in the results section since they are not experimental results but either included in material and methods section as info about the subjects of the study or as supplementary data.
- The authors address my comment about figure 2, however they completely remove it from the main document (to supplementary data). The gel present in figure 2 (first version was crowded with number making it impossible to visualize the protein spot and identify the number). However, this removes important information that is necessary to understand the results on up and downregulated proteins detected by the software used in their analysis. My suggestion of pin point some spot in the main document figure 2 was not provided by the authors referring that that is an image automatically generated by the software. I consider, that it is important the authors provide as figure 2 a gel representative (or a master gel) of the CF group and one representative (or a master gel) of non-CF group side by side (with the resolution as in previous figure 2 but without the numbers of the spots). The authors should have these as raw data gels. They can then (if not possible with the software) do the manual labelling the spots corresponding to the top 20 upregulated and 20 downregulated proteins as identified in table 2. This will make possible for the readers to observe on the gels the results shown in table 2.
- About comment 7, it is in fact normal that proteomics in 2D results show different spots identified as the same protein. However, 30 spots being the same protein (Alpha-2-macroglobulin) seems (to me) a high number to be justified with post translation modifications and putative random degradation. Can the authors comment on these specific high number of spots for one single protein? It is possible that a carryover phenomenon had happens in some of those spots? Could the authors provide the localization of these 30 spots in the gel as extra information just for the review, in order to understand where all these spots are localized in the gel/s?
- In table S2, there is one spot (668) with 2 proteins identified (Three prime repair exonuclease -Q9BQ50 and Tumor suppressor candidate 2 - O75896), however in table 2 there is only one protein referred for this spot. What is the correct info? Where there two proteins identified to this spot?
- It is not clear, from the methods description, how the first comparison analysis of the gels was done, since there were 19 gels, some with one CF and one non-CF samples and some with two CF samples. Did the 28 individuals CF gels analysis after normalization were compared to the 10 individual gels analysis after normalization? Or a master gel was created from the data of the 28 individual CF “gels” after normalization and the same was done for the 10 individual “gels” and then the 2 master gels (CF vs non-CF) were compared? Can the authors elaborate more about this first comparative analysis?
- Four and half pages of discussion it is to extensive I would suggest to the authors to summarize it to 2 and a half pages. Being to extensive makes the reader to lose interest.
Minor comments
- About my ER comment, the CFTR is degraded based the ER quality control, but not in the ER itself, the degradation takes place in the proteasome. So the sentence is not correct, stating that “….(ER), where it is subsequently degraded…” it is more correct if the sentence is altered to “….(ER), being subsequently degraded…”
- In Figure 3, A and B should be add to each panel in the figure. Figure 3A is too small, making it impossible to read some of the numbers present the figure. In figure 3B the CF individuals are identified as dark grey circles and ctlr as light grey circles in the legend. However, it seems to me that there are more light grey circles then dark grey circles in the graph and CF should be 28 and crtl 10 subject, or these groups are changed?
- In figure 5, the numbers inside the figure need to be alter since they are still 6A and 6B.

Author Response
The paper entitled “Serum-based proteomics profiling in adult patients with cystic fibrosis,” from Benabdelkamel H et al characterize the proteomic differences in plasma samples from patients with CF compared to a control group from subjects without CF.
This work represents an interesting aspect of CF research, usage of plasma samples from patients with CF and controls to discriminate by proteomics the proteins up and downregulated in the comparison between the two groups. They perform the proteomics analysis on the samples and then analyse the results using several bioinformatic tools. In summary, this paper brings improve of knowledge on the differences between CF and non-CF subject that may help find new biomarkers in the CF field, however needs major reformulation before be considered to publication. The research seems well constructed and some of my concerns are already addressed in the first review however there are still some issues that need to be clarified before the paper can be considered for publication.
Thank you again for going through our revised manuscript.
Major concerns
Comment 1 :From the info given by the authors to my first comment about mutations it is possible to infer that these results were not obtained during this study, just collected from the patient info by the clinician. Thus, sub-section 2.1 and table 1 should not be included in the results section since they are not experimental results but either included in material and methods section as info about the subjects of the study or as supplementary data.
Response: I understand the reviewer's perspective, however, this section was based on a clinical baseline questionnaire. What we presented in table 1 is a data collected by this questionnaire and we have reported in this first section of the results with some analysis to the cohort that we have. By definition this is a questionnaire based result.
Comment 2 :The authors address my comment about figure 2, however they completely remove it from the main document (to supplementary data). The gel present in figure 2 (first version was crowded with number making it impossible to visualize the protein spot and identify the number). However, this removes important information that is necessary to understand the results on up and downregulated proteins detected by the software used in their analysis. My suggestion of pin point some spot in the main document figure 2 was not provided by the authors referring that that is an image automatically generated by the software. I consider, that it is important the authors provide as figure 2 a gel representative (or a master gel) of the CF group and one representative (or a master gel) of non-CF group side by side (with the resolution as in previous figure 2 but without the numbers of the spots). The authors should have these as raw data gels. They can then (if not possible with the software) do the manual labelling the spots corresponding to the top 20 upregulated and 20 downregulated proteins as identified in table 2. This will make possible for the readers to observe on the gels the results shown in table 2.
Response: Thank you for your suggested regarding the Figure 2 (figure S1), now we provide with our best a new figure (with top 20 up regulated and 20 down regulated proteins using 2 image gels ( one from non-CF gels and one from CF gels) from raw data gels) as your suggest and we hope this time we clarified your concern.
Comment 3: About comment 7, it is in fact normal that proteomics in 2D results show different spots identified as the same protein. However, 30 spots being the same protein (Alpha-2-macroglobulin) seems (to me) a high number to be justified with post translation modifications and putative random degradation. Can the authors comment on these specific high number of spots for one single protein? It is possible that a carryover phenomenon had happens in some of those spots? Could the authors provide the localization of these 30 spots in the gel as extra information just for the review, in order to understand where all these spots are localized in the gel/s?
Response: We understand your concern regarding this issue but we will confirm you that our data is collected with high confidence and precision. In our case, as we explain before, we found several isoforms (PMTs) of the same proteins (expAlpha-2-macroglobulin), those Spots multiplicity is most likely a result phosphorylation, glycosylation, nitrosylation and oxidation, proteolytic cleavage and/or protein aggregation (also can happened ). MS analysis will yield similar peptides because proteins were likely trypsin-digested, and database search tools are set to look for proteolytic fragments specific to the enzyme (residues starting/ending with K and R).
Obviously there could be any number of reasons for the different spots as Glycosylation tends to produce horizontal (pI) movement (as in our case most of the spots are in the same horizontal line and closer each other’s number : 253,255,277,256,250,266,263,273 …ect please see the figure R1 only for your review) phosphorylation usually in both directions ( 77,184,149,291,289,362…..ect) and degradation obviously down, but it could be left or right, depending on the sequence.
And almost the ID proteins from those spots give us unique proteins (Alpha-2-macroglobulin) as a factor for confirmation by MALDI-TOF that those spots are (Alpha-2-macroglobulin) and no mixture proteins.( please see the raw data attached from MASCOT ).
In the previous study Chenggong Zong et al they identified over of 25 spots (PMTs) for the 20S complexes, indicating multiply modified subunits of cardiac proteasomes using 2D-Gel technology(1)
- Two-Dimensional Electrophoresis Based Characterization of Post-translational Modifications of Mammalian 20S Proteasome Complexes Chenggong Zong, Glen W Young, Yueju Wang, Haojie Lu, Ning Deng, Oliver Drews, Peipei Ping Proteomics. Author manuscript; available in PMC 2009 Dec 1. Published in final edited form as: Proteomics. 2008 Dec; 8(23-24): 5025–5037. doi: 10.1002/pmic.200800387 We hope now with explanations we satisfied your question. .
Figure R1 (For reviewer): Representative gel image depicting localization of Alpha-2-macroglobulin protein spots (see the attached)
Comment 4 :In table S2, there is one spot (668) with 2 proteins identified (Three prime repair exonuclease -Q9BQ50 and Tumor suppressor candidate 2 - O75896), however in table 2 there is only one protein referred for this spot. What is the correct info? Where there two proteins identified to this spot?
Response: Thank you for this excellent observation the correct one is (Three prime repair exonuclease -Q9BQ50) now the table S2 was corrected accordingly.
Comment 5 :It is not clear, from the methods description, how the first comparison analysis of the gels was done, since there were 19 gels, some with one CF and one non-CF samples and some with two CF samples. Did the 28 individuals CF gels analysis after normalization were compared to the 10 individual gels analysis after normalization? Or a master gel was created from the data of the 28 individual CF “gels” after normalization and the same was done for the 10 individual “gels” and then the 2 master gels (CF vs non-CF) were compared? Can the authors elaborate more about this first comparative analysis?
Response:as you mentioned we have 19 gels with 28 CF and 10 non-CF and each gels contented 2 samples some with one CF and one non-CF samples and some with two CF samples as the number not equals (please see the designee the experimental Supplementary Table S4).).
Regarding the normalization and the comparative analysis your second answer is the right one as the normalization done for all 28 CF gels and 10 non-CF gels with same normalization using the cy2 as internal standard (Cy2 gel used as an internal standard; this standard was normalized and matched across all gels), this section was already added by the authors in the first version as one of the reviewer ask for more details on it please see the methodology section (4.4 CyDye labeling, 2D-DIGE, and imaging)
As described here in more details “Differential in-gel electrophoresis (DIGE) images were analyzed using the Progenesis Same Spots v.3.3 software (Nonlinear Dynamics Ltd., UK). The gel images were first aligned together, and prominent spots were used to assign vectors to the digitized images within each gel manually. The automatic vector tool was next used to add additional vectors, which were manually revised and edited for correction if necessary. These vectors were used to warp and align gel images with a reference image of one internal standard across and within each gel. The gel groups were defined according to the experimental design, and the normalized volume of the spots was used to identify statistically significant differences. The software calculated the normalized volume of each spot on each gel from the Cy3 (or Cy5) to Cy2 spot volume ratio. The software performs a log transformation of the spot volumes to generate normally distributed data. The log normalized volume was used to quantify differential expression. Independent direct comparisons were made between the 28 CF patients and the 10 controls, and fold differences and p-values were calculated using a one-way ANOVA. All spots were pre-filtered and manually checked before applying the statistical criteria (ANOVA test, p ≤ 0.05, and fold difference ≥ 1.5). The normalized volume of spots, instead of spot intensities, was used in the statistical processing. Only those spots that fulfilled the above-mentioned statistical criteria were submitted for the MS analysis”
Comment 6Four and half pages of discussion it is to extensive I would suggest to the authors to summarize it to 2 and a half pages. Being to extensive makes the reader to lose interest.
Response: Based in the comments of the first 2 reviewers the discussion was increase as they suggested to added more paragraph regarding the biomarker and the network pathway additionally we added some of sentences as your requested in the discussion section so we believe if we summarize it more we will lose the order and the flow of discussion, so we will do our best to decrease it with one page as maximum and we hope you take it in the consideration.
Minor comments
Comment 7About my ER comment, the CFTR is degraded based the ER quality control, but not in the ER itself, the degradation takes place in the proteasome. So the sentence is not correct, stating that “….(ER), where it is subsequently degraded…” it is more correct if the sentence is altered to “….(ER), being subsequently degraded…”
Response:The sentence now corrected as you’re suggested.
Comment 8 In Figure 3, A and B should be add to each panel in the figure. Figure 3A is too small, making it impossible to read some of the numbers present the figure. In figure 3B the CF individuals are identified as dark grey circles and ctlr as light grey circles in the legend. However, it seems to me that there are more light grey circles then dark grey circles in the graph and CF should be 28 and crtl 10 subject, or these groups are changed?
Response: It seems there are several spots under each others, we have regenerated the figures from different angle (different PC) and in grey (for review) and colorful for publication.
Comment 9In figure 5, the numbers inside the figure need to be alter since they are still 6A and 6B.
Response: Thank you, the numbers now corrected.

Round 3
Reviewer 2 Report
The authors have done a good effort to update the manuscripts and reply to my last comments.
I think the manuscript can be now considered for publication.
This manuscript is a resubmission of an earlier submission. The following is a list of the peer review reports and author responses from that submission.
Round 1
Reviewer 1 Report
In this manuscript, Benabdelkamel et al aim at using serum proteomics to identify novel biomarkers for Cystic Fibrosis. Although the subject is relevant, the study is very preliminary and has serious flaws.
Major aspects
- There is no mention to the number of technical and biological replicates used for the 2D-based MS protein identification. All the descriptions suggest that it was done only once for each patient. It is also difficult to understand how statistical analysis of the results was performed.
- There is no discussion on the possible bias introduced by the use of different genotypes – furthermore, in the submitted version, this is even difficult to understand as although a Table with individual genotypes is provided, there is no summary for the genotypes used.
- The study includes no validation – at least some of the differentially expressed proteins identified would need to be validated by a different method (e.g.WB).
- The bioinformatics analysis is provided without extracting any biological relevance from it.
- The authors mention that they identified 134 differentially expressed proteins. Later, this number decreased to 59, without any explanation being provided.
- I1234V is mainly a splicing mutation – thus supposedly belonging to class V and not class IV.
- 5 is unacceptable as a figure in a publication – the work is not on the development of a novel bioinformatics analysis tool, so “print screen” of the results obtained using an online tool is not valid as a figure.
Minor aspects
- Use of English is poor…
- In the introduction, the authors should try to be more.
- The quality of the figures is very low.
Reviewer 2 Report
This is an interesting article that focuses on an issue of great relevance in the comprehensive searching for potential novel biomarkers of Cystic Fibrosis such as therapeutic targets, evolution of the diseases and comorbilities, etc. However, it requires an in-depth review and the contribution of result that certainly have and have not been considered to know:
- Table 1 with the description of study population is very limited. They have to include the clinical parameters of the patients: evolution time, comorbilities (diabetes, exocrine fails, infections diseases, etc); the biochemical data of patients and controls (lipids, glucose, etc..)
- In the introduction is missing many relevant literature.
- The quality of figures have to improve.
- They do not identify clearly the potential biomarkers and it is missing the ROC curve.
- The CF is a conformational disease, and it is very important to identify by proteomics the unfolding state of the proteins. Do they have any data in this topic?
- The discussion have to include which proteins could be biomarkers y the analysis of the ROC curve and the correlation with the clinical data.